# Reducing Reparameterization Gradient Variance

**Andrew C. Miller**[*]
Harvard University
acm@seas.harvard.edu

**Nicholas J. Foti**
University of Washington
nfoti@uw.edu

**Alexander D'Amour**
UC Berkeley
alexdamour@berkeley.edu

**Ryan P. Adams**
Google Brain and Princeton University
rpa@princeton.edu

## Abstract

Optimization with noisy gradients has become ubiquitous in statistics and machine learning. Reparameterization gradients, or gradient estimates computed via the "reparameterization trick," represent a class of noisy gradients often used in Monte Carlo variational inference (MCVI). However, when these gradient estimators are too noisy, the optimization procedure can be slow or fail to converge. One way to reduce noise is to generate more samples for the gradient estimate, but this can be computationally expensive. Instead, we view the noisy gradient as a random variable, and form an inexpensive approximation of the generating procedure for the gradient sample. This approximation has high correlation with the noisy gradient by construction, making it a useful control variate for variance reduction. We demonstrate our approach on a non-conjugate hierarchical model and a Bayesian neural net where our method attained orders of magnitude (20-2,000$\times$) reduction in gradient variance resulting in faster and more stable optimization.

## 1   Introduction

Representing massive datasets with flexible probabilistic models has been central to the success of many statistics and machine learning applications, but the computational burden of fitting these models is a major hurdle. For optimization-based fitting methods, a central approach to this problem has been replacing expensive evaluations of the gradient of the objective function with cheap, unbiased, stochastic estimates of the gradient. For example, stochastic gradient descent using small minibatches of (conditionally) i.i.d. data to estimate the gradient at each iteration is a popular approach with massive data sets. Alternatively, some learning methods sample directly from a generative model or approximating distribution to estimate the gradients of interest, for example, in learning algorithms for implicit models [18, 30] and generative adversarial networks [2, 9].

Approximate Bayesian inference using variational techniques (variational inference, or VI) has also motivated the development of new stochastic gradient estimators, as the variational approach reframes the integration problem of inference as an optimization problem [4]. VI approaches seek out the distribution from a well-understood variational family of distributions that best approximates an intractable posterior distribution. The VI objective function itself is often intractable, but recent work has shown that it can be optimized with stochastic gradient methods that use Monte Carlo estimates of the gradient [19, 14, 22, 25], which we call Monte Carlo variational inference (MCVI). In MCVI, generating samples from an approximate posterior distribution is the source of gradient stochasticity. Alternatively, *stochastic variational inference* (SVI) [11] and other stochastic opti-

---

[*] http://andymiller.github.io/

mization procedures induce stochasticity through data subsampling; MCVI can also be augmented with data subsampling to accelerate computation for large data sets.

The two commonly used MCVI gradient estimators are the *score function gradient* [19, 22] and the *reparameterization gradient* [14, 25, 29, 8]. Broadly speaking, score function estimates can be applied to both discrete and continuous variables, but often have high variance and thus are frequently used in conjunction with variance reduction techniques. On the other hand, the reparameterization gradient often has lower variance, but is restricted to continuous random variables. See Ruiz et al. [28] for a unifying perspective on these two estimators. Like other stochastic gradient methods, the success of MCVI depends on controlling the variance of the stochastic gradient estimator.

In this work, we present a novel approach to controlling the variance of the *reparameterization gradient estimator* in MCVI. Existing MCVI methods control this variance naïvely by averaging several gradient estimates, which becomes expensive for large data sets and complex models, with error that only diminishes as $O(1/\sqrt{N})$. Our approach exploits the fact that, in MCVI, the randomness in the gradient estimator is completely determined by a known Monte Carlo generating process; this allows us to leverage knowledge about this generative procedure to de-noise the gradient estimator. In particular, we construct a computationally cheap control variate based on an analytical linear approximation to the gradient estimator. Taking a linear combination of a naïve gradient estimate with this control variate yields a new estimator for the gradient that remains unbiased but has lower variance. Applying the idea to Gaussian approximating families, we observe a 20-2,000× reduction in variance of the gradient norm under various conditions, and faster convergence and more stable behavior of optimization traces.

## 2   Background

**Variational Inference**   Given a model, $p(\boldsymbol{z}, \mathcal{D}) = p(\mathcal{D}|\boldsymbol{z})p(\boldsymbol{z})$, of data $\mathcal{D}$ and parameters/latent variables $\boldsymbol{z}$, the goal of VI is to approximate the posterior distribution $p(\boldsymbol{z}|\mathcal{D})$. VI approximates this intractable posterior distribution with one from a simpler family, $\mathcal{Q} = \{q(\boldsymbol{z}; \boldsymbol{\lambda}), \boldsymbol{\lambda} \in \boldsymbol{\Lambda}\}$, parameterized by *variational parameters* $\boldsymbol{\lambda}$. VI procedures seek out the member of that family, $q(\cdot; \boldsymbol{\lambda}) \in \mathcal{Q}$, that minimizes some divergence between the approximation $q$ and the true posterior $p(\boldsymbol{z}|\mathcal{D})$.

Variational inference can be framed as an optimization problem, usually in terms of Kullback-Leibler (KL) divergence, of the following form

$$\boldsymbol{\lambda}^* = \underset{\boldsymbol{\lambda} \in \Lambda}{\arg\min}\, \mathrm{KL}(q(\boldsymbol{z}; \boldsymbol{\lambda}) \,||\, p(\boldsymbol{z}|\mathcal{D})) = \underset{\boldsymbol{\lambda} \in \Lambda}{\arg\min}\, \mathbb{E}_{\boldsymbol{z} \sim q_{\boldsymbol{\lambda}}}\left[\ln q(\boldsymbol{z}; \boldsymbol{\lambda}) - \ln p(\boldsymbol{z}|\mathcal{D})\right] .$$

The task is to find a setting of $\boldsymbol{\lambda}$ that makes $q(\boldsymbol{z}; \boldsymbol{\lambda})$ close to the posterior $p(\boldsymbol{z}|\mathcal{D})$ in KL divergence.[2] Directly computing the KL divergence requires evaluating the posterior itself; therefore, VI procedures use the *evidence lower bound* (ELBO) as the optimization objective

$$\mathcal{L}(\boldsymbol{\lambda}) = \mathbb{E}_{\boldsymbol{z} \sim q_{\boldsymbol{\lambda}}}\left[\ln p(\boldsymbol{z}, \mathcal{D}) - \ln q(\boldsymbol{z}; \boldsymbol{\lambda})\right], \tag{1}$$

which, when maximized, minimizes the KL divergence between $q(\boldsymbol{z}; \boldsymbol{\lambda})$ and $p(\boldsymbol{z}|\mathcal{D})$. In special cases, parts of the ELBO can be expressed analytically (e.g. the entropy form or KL-to-prior form [10]) — we focus on the general form in Equation 1.

To maximize the ELBO with gradient methods, we need to compute the gradient of Eq. (1), $\partial\mathcal{L}/\partial\boldsymbol{\lambda} \triangleq \boldsymbol{g}_{\boldsymbol{\lambda}}$. The gradient inherits the ELBO's form as an expectation, which is in general an intractable quantity to compute. In this work, we focus on *reparameterization gradient estimators* (RGEs) computed using the *reparameterization trick*. The reparameterization trick exploits the structure of the *variational data generating procedure* — the mechanism by which $\boldsymbol{z}$ is simulated from $q_{\boldsymbol{\lambda}}(\boldsymbol{z})$. To compute the RGE, we first express the sampling procedure from $q_{\boldsymbol{\lambda}}(\boldsymbol{z})$ as a differentiable map applied to exogenous randomness

$$\epsilon \sim q_0(\epsilon) \qquad\qquad\qquad \text{independent of } \boldsymbol{\lambda} \tag{2}$$
$$\boldsymbol{z} = \mathcal{T}(\epsilon; \boldsymbol{\lambda}) \qquad\qquad\qquad \text{differentiable map,} \tag{3}$$

where the initial distribution $q_0$ and $\mathcal{T}$ are jointly defined such that $\boldsymbol{z} \sim q(\boldsymbol{z}; \boldsymbol{\lambda})$ has the desired distribution. As a simple concrete example, if we set $q(\boldsymbol{z}; \boldsymbol{\lambda})$ to be a diagonal Gaussian,

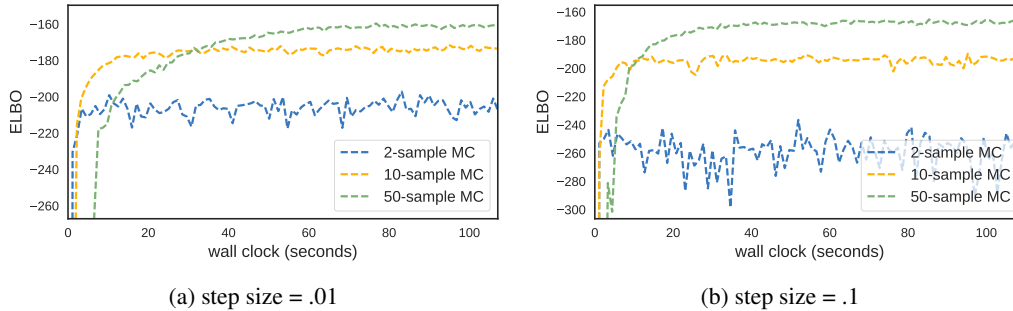

(a) step size = .01                    (b) step size = .1

Figure 1: Optimization traces for MCVI applied to a Bayesian neural network with various hyper-parameter settings. Each trace is running `adam` [13]. The three lines in each plot correspond to three different numbers of samples, $L$, used to estimate the gradient at each step. (Left) small step-size; (Right) stepsize 10 times larger. Large step sizes allow for quicker progress, however noisier (i.e., small $L$) gradients combined with large step sizes result in chaotic optimization dynamics. The converging traces reach different ELBOs due to the illustrative constant learning rates; in practice, one decreases the step size over time to satisfy the convergence criteria in Robbins and Monro [26].

$\mathcal{N}(\boldsymbol{m_\lambda}, \mathrm{diag}(\boldsymbol{s_\lambda^2}))$, with $\boldsymbol{\lambda} = [\boldsymbol{m_\lambda}, \boldsymbol{s_\lambda}]$, $\boldsymbol{m_\lambda} \in \mathbb{R}^D$, and $\boldsymbol{s_\lambda} \in \mathbb{R}^D_+$ the mean and variance. The sampling procedure could then be defined as

$$\epsilon \sim \mathcal{N}(0, I_D), \qquad \boldsymbol{z} = \mathcal{T}(\epsilon; \boldsymbol{\lambda}) = \boldsymbol{m_\lambda} + \boldsymbol{s_\lambda} \odot \epsilon, \tag{4}$$

where $\boldsymbol{s} \odot \epsilon$ denotes an element-wise product.[3] Given this map, the reparameterization gradient estimator is simply the gradient of a Monte Carlo ELBO estimate with respect to $\boldsymbol{\lambda}$. For a single sample, this is

$$\epsilon \sim q_0(\epsilon), \qquad \boldsymbol{\hat{g}_\lambda} \triangleq \nabla_{\boldsymbol{\lambda}} \left[ \ln p(\mathcal{T}(\epsilon; \boldsymbol{\lambda}), \mathcal{D}) - \ln q(\mathcal{T}(\epsilon; \boldsymbol{\lambda}); \boldsymbol{\lambda}) \right]$$

and similarly the $L$-sample approximation can be computed by averaging the single-sample estimator over the individual samples

$$\boldsymbol{\hat{g}_\lambda}^{(L)} = \frac{1}{L} \sum_{\ell=1}^{L} \boldsymbol{\hat{g}_\lambda}(\epsilon^\ell). \tag{5}$$

Crucially, the reparameterization gradient is unbiased, $\mathbb{E}[\boldsymbol{\hat{g}_\lambda}] = \nabla_{\boldsymbol{\lambda}} \mathcal{L}(\boldsymbol{\lambda})$, guaranteeing the convergence of stochastic gradient optimization procedures that use it [26].

**Gradient Variance and Convergence**  The efficiency of Monte Carlo variational inference hinges on the magnitude of gradient noise and the step size chosen for the optimization procedure. When the gradient noise is large, smaller gradient steps must be taken to avoid unstable dynamics of the iterates. However, a smaller step size increases the number of iterations that must be performed to reach convergence.

We illustrate this trade-off in Figure 1, which shows realizations of an optimization procedure applied to a Bayesian neural network using reparameterization gradients. The posterior is over $D = 653$ parameters that we approximate with a diagonal Gaussian (see Appendix C.2). We compare the progress of the `adam` algorithm using various numbers of samples [13], fixing the learning rate. The noise present in the single-sample estimator causes extremely slow convergence, whereas the lower noise 50-sample estimator quickly converges, albeit at 50 times the cost.

The upshot is that with low noise gradients we are able to safely take larger steps, enabling faster convergence to a local optimum. A natural question is, how can we reduce the variance of gradient estimates without introducing too much extra computation? Our approach is to use information about the variational model, $q(\cdot; \boldsymbol{\lambda})$, and carefully construct a control variate to the gradient.

**Control Variates**  Control variates are random quantities that are used to reduce the variance of a statistical estimator without introducing any bias by incorporating additional information into the estimator, [7]. Given an unbiased estimator $\boldsymbol{\hat{g}}$ such that $\mathbb{E}[\boldsymbol{\hat{g}}] = \boldsymbol{g}$ (the quantity of interest), our goal

is to construct another unbiased estimator with lower variance. We can do this by defining a *control variate* $\tilde{g}$ with *known expectation* $\tilde{m}$ and can write the new estimator as

$$\boldsymbol{g}^{(cv)} = \hat{\boldsymbol{g}} - \boldsymbol{C}(\tilde{\boldsymbol{g}} - \tilde{\boldsymbol{m}}) \,. \tag{6}$$

where $\boldsymbol{C} \in \mathbb{R}^{D \times D}$ for $D$-dimensional $\hat{\boldsymbol{g}}$. Clearly the new estimator has the same expectation as the original estimator, but has a different variance. We can attain optimal variance reduction by appropriately setting $\boldsymbol{C}$. Intuitively, the optimal $\boldsymbol{C}$ is very similar to a regression coefficient — it is related to the *covariance* between the control variate and the original estimator. See Appendix A for further details on optimally setting $\boldsymbol{C}$.

## 3   Method: Modeling Reparameterization Gradients

In this section we develop our main contribution, a new gradient estimator that can dramatically reduce reparameterization gradient variance. In MCVI, the reparameterization gradient estimator (RGE) is a Monte Carlo estimator of the true gradient — the estimator itself is a random variable. This random variable is generated using the "reparameterization trick" — we first generate some randomness $\epsilon$ and then compute the gradient of the ELBO with respect to $\boldsymbol{\lambda}$ holding $\epsilon$ fixed. This results in a complex distribution from which we can generate samples, but in general cannot characterize due to the complexity of the term arising from the gradient of the model term.

However, we do have a lot of information about the sampling procedure — we know the variational distribution $\ln q(\boldsymbol{z}; \boldsymbol{\lambda})$, the transformation $\mathcal{T}$, and we can evaluate the model joint density $\ln p(\boldsymbol{z}, \mathcal{D})$ pointwise. Furthermore, with automatic differentiation, it is often straightforward to obtain gradients and Hessian-vector products of our model $\ln p(\boldsymbol{z}, \mathcal{D})$. We propose a scheme that uses the structure of $q_{\boldsymbol{\lambda}}$ and curvature of $\ln p(\boldsymbol{z}, \mathcal{D})$ to construct a tractable approximation of the distribution of the RGE.[4] This approximation has a known mean and is correlated with the RGE distribution, allowing us to use it as a control variate to reduce the RGE variance.

Given a variational family parameterized by $\boldsymbol{\lambda}$, we can decompose the ELBO gradient into a few terms that reveal its "data generating procedure"

$$\epsilon \sim q_0 \,, \quad \boldsymbol{z} = \mathcal{T}(\epsilon; \boldsymbol{\lambda}) \tag{7}$$

$$\hat{\boldsymbol{g}}_{\boldsymbol{\lambda}} \triangleq \hat{\boldsymbol{g}}(\boldsymbol{z}; \boldsymbol{\lambda}) = \underbrace{\frac{\partial \ln p(\boldsymbol{z}, \mathcal{D})}{\partial \boldsymbol{z}}}_{\text{data term}} \frac{\partial \boldsymbol{z}}{\partial \boldsymbol{\lambda}} - \underbrace{\frac{\partial \ln q_{\boldsymbol{\lambda}}(\boldsymbol{z})}{\partial \boldsymbol{z}}}_{\text{pathwise score}} \frac{\partial \boldsymbol{z}}{\partial \boldsymbol{\lambda}} - \underbrace{\frac{\partial \ln q_{\boldsymbol{\lambda}}(\boldsymbol{z})}{\partial \boldsymbol{\lambda}}}_{\text{parameter score}} \,. \tag{8}$$

Certain terms in Eq. (8) have tractable distributions. The Jacobian of $\mathcal{T}(\cdot; \boldsymbol{\lambda})$, given by $\partial \boldsymbol{z}/\partial \boldsymbol{\lambda}$, is defined by our choice of $q(\boldsymbol{z}; \boldsymbol{\lambda})$. For some transformations $\mathcal{T}$ we can exactly compute the distribution of the Jacobian given the distribution of $\epsilon$. The *pathwise* and *parameter score* terms are gradients of our approximate distribution with respect to $\boldsymbol{\lambda}$ (via $\boldsymbol{z}$ or directly). If our approximation is tractable (e.g., a multivariate Gaussian), we can exactly characterize the distribution for these components.[5]

However, the *data term* in Eq. (8) involves a potentially complicated function of the latent variable $\boldsymbol{z}$ (and therefore a complicated function of $\epsilon$), resulting in a difficult-to-characterize distribution. Our goal is to construct an approximation to the distribution of $\partial \ln p(\boldsymbol{z}, \mathcal{D})/\partial \boldsymbol{z}$ and its interaction with $\partial \boldsymbol{z}/\partial \boldsymbol{\lambda}$ given a fixed distribution over $\epsilon$. If the approximation yields random variables that are highly correlated with $\hat{\boldsymbol{g}}_{\boldsymbol{\lambda}}$, then we can use it to reduce the variance of that RGE sample.

**Linearizing the data term**   To simplify notation, we write the data term of the gradient as

$$\boldsymbol{f}(\boldsymbol{z}') \triangleq \left. \frac{\partial \ln p(\boldsymbol{z}, \mathcal{D})}{\partial \boldsymbol{z}} \right|_{\boldsymbol{z}=\boldsymbol{z}'} \,, \tag{9}$$

where $\boldsymbol{f} : \mathbb{R}^D \mapsto \mathbb{R}^D$ since $\boldsymbol{z} \in \mathbb{R}^D$. We then linearize $\boldsymbol{f}$ about some value $\boldsymbol{z}_0$

$$\tilde{\boldsymbol{f}}(\boldsymbol{z}) = \boldsymbol{f}(\boldsymbol{z}_0) + \left[ \frac{\partial \boldsymbol{f}}{\partial \boldsymbol{z}}(\boldsymbol{z}_0) \right] (\boldsymbol{z} - \boldsymbol{z}_0) = \boldsymbol{f}(\boldsymbol{z}_0) + \boldsymbol{H}(\boldsymbol{z}_0)(\boldsymbol{z} - \boldsymbol{z}_0), \tag{10}$$

where $\boldsymbol{H}(\boldsymbol{z}_0)$ is the Hessian of the model, $\ln p(\boldsymbol{z}, \mathcal{D})$, with respect to $\boldsymbol{z}$ evaluated at $\boldsymbol{z}_0$,

$$\boldsymbol{H}(\boldsymbol{z}_0) = \frac{\partial \boldsymbol{f}}{\partial \boldsymbol{z}}(\boldsymbol{z}_0) = \frac{\partial^2 \ln p(\boldsymbol{z}, \mathcal{D})}{\partial \boldsymbol{z}^2}(\boldsymbol{z}_0) \tag{11}$$

Note that even though this uses second-order information about the model, it is a first-order approximation of the gradient. We also view this as a transformation of the random $\epsilon$ for a fixed $\boldsymbol{\lambda}$

$$\tilde{\boldsymbol{f}}_{\boldsymbol{\lambda}}(\epsilon) = \boldsymbol{f}(\boldsymbol{z}_0) + \boldsymbol{H}(\boldsymbol{z}_0)(\mathcal{T}(\epsilon, \boldsymbol{\lambda}) - \boldsymbol{z}_0)\,, \tag{12}$$

which is linear in $\boldsymbol{z} = \mathcal{T}(\epsilon, \boldsymbol{\lambda})$. For some forms of $\mathcal{T}$ we can analytically derive the distribution of the random variable $\tilde{\boldsymbol{f}}_{\boldsymbol{\lambda}}(\epsilon)$. In Eq. (8), the *data term* interacts with the Jacobian of $\mathcal{T}$, given by

$$\boldsymbol{J}_{\boldsymbol{\lambda}'}(\epsilon) \triangleq \frac{\partial \boldsymbol{z}}{\partial \boldsymbol{\lambda}} = \frac{\partial \mathcal{T}(\epsilon, \boldsymbol{\lambda})}{\partial \boldsymbol{\lambda}}\bigg|_{\boldsymbol{\lambda} = \boldsymbol{\lambda}'}\,, \tag{13}$$

which importantly is a function of the same $\epsilon$ as in Eq. (12). We form our approximation of the first term in Eq. (8) by multiplying Eqs. (12) and (13) yielding

$$\tilde{\boldsymbol{g}}_{\boldsymbol{\lambda}}^{(data)}(\epsilon) \triangleq \tilde{\boldsymbol{f}}_{\boldsymbol{\lambda}}(\epsilon)\boldsymbol{J}_{\boldsymbol{\lambda}}(\epsilon)\,. \tag{14}$$

The tractability of this approximation hinges on how Eq. (14) depends on $\epsilon$. When $q(\boldsymbol{z}; \boldsymbol{\lambda})$ is multivariate normal, we show that this approximation has a computable mean and can be used to reduce variance in MCVI settings. In the following sections we describe and empirically test this variance reduction technique applied to diagonal Gaussian posterior approximations.

## 3.1 Gaussian Variational Families

Perhaps the most common choice of approximating distribution for MCVI is a diagonal Gaussian, parameterized by a mean $\boldsymbol{m}_{\boldsymbol{\lambda}} \in \mathbb{R}^D$ and scales $\boldsymbol{s}_{\boldsymbol{\lambda}} \in \mathbb{R}_+^D$. [6] The log probability density function is

$$\ln q(\boldsymbol{z}; \boldsymbol{m}_{\boldsymbol{\lambda}}, \boldsymbol{s}_{\boldsymbol{\lambda}}^2) = -\frac{1}{2}(\boldsymbol{z} - \boldsymbol{m}_{\boldsymbol{\lambda}})^{\mathsf{T}}\left[\mathrm{diag}(\boldsymbol{s}_{\boldsymbol{\lambda}}^2)\right]^{-1}(\boldsymbol{z} - \boldsymbol{m}_{\boldsymbol{\lambda}}) - \frac{1}{2}\sum_d \ln s_{\boldsymbol{\lambda}, d}^2 - \frac{D}{2}\ln(2\pi)\,. \tag{15}$$

To generate a random variate $\boldsymbol{z}$ from this distribution, we use the sampling procedure in Eq. (4). We denote the Monte Carlo RGE as $\hat{\boldsymbol{g}}_{\boldsymbol{\lambda}} \triangleq [\hat{\boldsymbol{g}}_{\boldsymbol{m}_{\boldsymbol{\lambda}}}, \hat{\boldsymbol{g}}_{\boldsymbol{s}_{\boldsymbol{\lambda}}}]$. From Eq. (15), it is straightforward to derive the distributions of the *pathwise score*, *parameter score*, and *Jacobian* terms in Eq. (8).

The *Jacobian* term of the sampling procedure has two straightforward components

$$\frac{\partial \boldsymbol{z}}{\partial \boldsymbol{m}_{\boldsymbol{\lambda}}} = I_D\,, \quad \frac{\partial \boldsymbol{z}}{\partial \boldsymbol{s}_{\boldsymbol{\lambda}}} = \mathrm{diag}(\epsilon)\,. \tag{16}$$

The *pathwise score* term is the partial derivative of Eq. (15) with respect to $\boldsymbol{z}$, ignoring variation due to the variational distribution parameters and noting that $\boldsymbol{z} = \boldsymbol{m}_{\boldsymbol{\lambda}} + \boldsymbol{s}_{\boldsymbol{\lambda}} \odot \epsilon$:

$$\frac{\partial \ln q}{\partial \boldsymbol{z}} = -\mathrm{diag}(\boldsymbol{s}_{\boldsymbol{\lambda}}^2)^{-1}(\boldsymbol{z} - \boldsymbol{m}_{\boldsymbol{\lambda}}) = -\epsilon/\boldsymbol{s}_{\boldsymbol{\lambda}}\,. \tag{17}$$

The *parameter score* term is the partial derivative of Eq. (15) with respect to variational parameters $\boldsymbol{\lambda}$, ignoring variation due to $\boldsymbol{z}$. The $\boldsymbol{m}_{\boldsymbol{\lambda}}$ and $\boldsymbol{s}_{\boldsymbol{\lambda}}$ components are given by

$$\frac{\partial \ln q}{\partial \boldsymbol{m}_{\boldsymbol{\lambda}}} = (\boldsymbol{z} - \boldsymbol{m}_{\boldsymbol{\lambda}})/s_{\boldsymbol{\lambda}}^2 = \epsilon/\boldsymbol{s}_{\boldsymbol{\lambda}} \tag{18}$$

$$\frac{\partial \ln q}{\partial \boldsymbol{s}_{\boldsymbol{\lambda}}} = -1/\boldsymbol{s}_{\boldsymbol{\lambda}} - (\boldsymbol{z} - \boldsymbol{m}_{\boldsymbol{\lambda}})^2/s_{\boldsymbol{\lambda}}^2 = \frac{\epsilon^2 - 1}{\boldsymbol{s}_{\boldsymbol{\lambda}}}\,. \tag{19}$$

The *data term*, $\boldsymbol{f}(\boldsymbol{z})$, multiplied by the Jacobian of $\mathcal{T}$ is all that remains to be approximated in Eq. (8). We linearize $\boldsymbol{f}$ around $\boldsymbol{z}_0 = \boldsymbol{m}_{\boldsymbol{\lambda}}$ where the approximation is expected to be accurate

$$\tilde{\boldsymbol{f}}_{\boldsymbol{\lambda}}(\epsilon) = \boldsymbol{f}(\boldsymbol{m}_{\boldsymbol{\lambda}}) + \boldsymbol{H}(\boldsymbol{m}_{\boldsymbol{\lambda}})\left((\boldsymbol{m}_{\boldsymbol{\lambda}} + \boldsymbol{s}_{\boldsymbol{\lambda}} \odot \epsilon) - \boldsymbol{m}_{\boldsymbol{\lambda}}\right) \tag{20}$$

$$\sim \mathcal{N}\left(\boldsymbol{f}(\boldsymbol{m}_{\boldsymbol{\lambda}}), \boldsymbol{H}(\boldsymbol{m}_{\boldsymbol{\lambda}})\mathrm{diag}(\boldsymbol{s}_{\boldsymbol{\lambda}}^2)\boldsymbol{H}(\boldsymbol{m}_{\boldsymbol{\lambda}})^{\mathsf{T}}\right)\,. \tag{21}$$

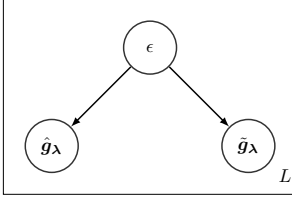

Figure 2: Relationship between the base randomness $\epsilon$, RGE $\hat{g}$, and approximation $\tilde{g}$. Arrows indicate deterministic functions. Sharing $\epsilon$ correlates the random variables. We know the distribution of $\tilde{g}$, which allows us to use it as a control variate for $\hat{g}$.

**Algorithm 1** Gradient descent with RV-RGE with a diagonal Gaussian variational family

1: **procedure** RV-RGE-OPTIMIZE($\boldsymbol{\lambda}_1, \ln p(\boldsymbol{z}, \mathcal{D}), L$)
2:     $\boldsymbol{f}(\boldsymbol{z}) \leftarrow \nabla_{\boldsymbol{z}} \ln p(\boldsymbol{z}, \mathcal{D})$
3:     $\boldsymbol{H}(\boldsymbol{z}_a, \boldsymbol{z}_b) \leftarrow \left[ \nabla_{\boldsymbol{z}}^2 \ln p(\boldsymbol{z}_a, \mathcal{D}) \right] \boldsymbol{z}_b$      ▷ Define Hessian-vector product function
4:     **for** $t = 1, \ldots, T$ **do**
5:        $\epsilon^{(\ell)} \sim \mathcal{N}(0, I_D)$ for $\ell = 1, \ldots, L$      ▷ Base randomness $q_0$
6:        $\hat{\boldsymbol{g}}_{\boldsymbol{\lambda}_t}^{(\ell)} \leftarrow \nabla_{\boldsymbol{\lambda}} \ln p(\boldsymbol{z}(\epsilon^{(\ell)}, \boldsymbol{\lambda}_t), \mathcal{D})$      ▷ Reparameterization gradients
7:        $\tilde{\boldsymbol{g}}_{m_{\boldsymbol{\lambda}_t}}^{(\ell)} \leftarrow \boldsymbol{f}(\boldsymbol{m}_{\boldsymbol{\lambda}_t}) + \boldsymbol{H}(\boldsymbol{m}_{\boldsymbol{\lambda}_t}, \boldsymbol{s}_{\boldsymbol{\lambda}_t} \odot \epsilon^{(\ell)})$      ▷ Mean approx
8:        $\tilde{\boldsymbol{g}}_{s_{\boldsymbol{\lambda}_t}}^{(\ell)} \leftarrow \left( \boldsymbol{f}(\boldsymbol{m}_{\boldsymbol{\lambda}_t}) + \boldsymbol{H}(\boldsymbol{m}_{\boldsymbol{\lambda}_t}, \boldsymbol{s}_{\boldsymbol{\lambda}_t} \odot \epsilon^{(\ell)}) \right) \odot \epsilon + \frac{1}{\boldsymbol{s}_{\boldsymbol{\lambda}_t}}$      ▷ Scale approx
9:        $\mathbb{E}[\tilde{\boldsymbol{g}}_{m_{\boldsymbol{\lambda}_t}}] \leftarrow \boldsymbol{f}(\boldsymbol{m}_{\boldsymbol{\lambda}_t})$      ▷ Mean approx expectation
10:       $\mathbb{E}[\tilde{\boldsymbol{g}}_{s_{\boldsymbol{\lambda}_t}}] \leftarrow \mathrm{diag}(\boldsymbol{H}(\boldsymbol{m}_{\boldsymbol{\lambda}_t})) \odot \boldsymbol{s}_{\boldsymbol{\lambda}_t} + 1/\boldsymbol{s}_{\boldsymbol{\lambda}_t}$      ▷ Scale approx expectation
11:       $\hat{\boldsymbol{g}}_{\boldsymbol{\lambda}_t}^{(RV)} = \frac{1}{L} \sum_{\ell} \hat{\boldsymbol{g}}_{\boldsymbol{\lambda}_t}^{\ell} - (\tilde{\boldsymbol{g}}_{\boldsymbol{\lambda}_t}^{\ell} - \mathbb{E}[\tilde{\boldsymbol{g}}_{\boldsymbol{\lambda}_t}])$      ▷ Subtract control variate
12:       $\boldsymbol{\lambda}_{t+1} \leftarrow \texttt{grad-update}(\boldsymbol{\lambda}_t, \hat{\boldsymbol{g}}_{\boldsymbol{\lambda}_t}^{(RV)})$      ▷ Gradient step (sgd, adam, etc.)
13:     **return** $\boldsymbol{\lambda}_T$

**Putting It Together: Full RGE Approximation** We write the complete approximation of the RGE in Eq. (8) by combining Eqs. (16), (17), (18), (19), and (21) which results in two components that are concatenated, $\tilde{\boldsymbol{g}}_{\boldsymbol{\lambda}} = [\tilde{\boldsymbol{g}}_{m_{\boldsymbol{\lambda}}}, \tilde{\boldsymbol{g}}_{s_{\boldsymbol{\lambda}}}]$. Each component is defined as

$$\tilde{\boldsymbol{g}}_{m_{\boldsymbol{\lambda}}} = \tilde{\boldsymbol{f}}_{\boldsymbol{\lambda}}(\epsilon) + \epsilon/\boldsymbol{s}_{\boldsymbol{\lambda}} - \epsilon/\boldsymbol{s}_{\boldsymbol{\lambda}} \qquad\qquad\qquad = \boldsymbol{f}(\boldsymbol{m}_{\boldsymbol{\lambda}}) + \boldsymbol{H}(\boldsymbol{m}_{\boldsymbol{\lambda}})(\boldsymbol{s}_{\boldsymbol{\lambda}} \odot \epsilon) \tag{22}$$

$$\tilde{\boldsymbol{g}}_{s_{\boldsymbol{\lambda}}} = \tilde{\boldsymbol{f}}_{\boldsymbol{\lambda}}(\epsilon) \odot \epsilon + (\epsilon/\boldsymbol{s}_{\boldsymbol{\lambda}}) \odot \epsilon - \frac{\epsilon^2 - 1}{\boldsymbol{s}_{\boldsymbol{\lambda}}} \quad = (\boldsymbol{f}(\boldsymbol{m}_{\boldsymbol{\lambda}}) + \boldsymbol{H}(\boldsymbol{m}_{\boldsymbol{\lambda}})(\boldsymbol{s}_{\boldsymbol{\lambda}} \odot \epsilon)) \odot \epsilon + \frac{1}{\boldsymbol{s}_{\boldsymbol{\lambda}}}. \tag{23}$$

To summarize, we have constructed an approximation, $\tilde{\boldsymbol{g}}_{\boldsymbol{\lambda}}$, of the reparameterization gradient, $\hat{\boldsymbol{g}}_{\boldsymbol{\lambda}}$, as a function of $\epsilon$. Because both $\tilde{\boldsymbol{g}}_{\boldsymbol{\lambda}}$ and $\hat{\boldsymbol{g}}_{\boldsymbol{\lambda}}$ are functions of the same random variable $\epsilon$, and because we have mimicked the random process that generates true gradient samples, the two gradient estimators will be correlated. This approximation yields two tractable distributions — a Gaussian for the mean parameter gradient, $\boldsymbol{g}_{m_{\boldsymbol{\lambda}}}$, and a location shifted, scaled non-central $\chi^2$ for the scale parameter gradient $\boldsymbol{g}_{s_{\boldsymbol{\lambda}}}$. Importantly, we can compute the mean of each component

$$\mathbb{E}[\tilde{\boldsymbol{g}}_{m_{\boldsymbol{\lambda}}}] = \boldsymbol{f}(\boldsymbol{m}_{\boldsymbol{\lambda}}), \qquad \mathbb{E}[\tilde{\boldsymbol{g}}_{s_{\boldsymbol{\lambda}}}] = \mathrm{diag}(\boldsymbol{H}(\boldsymbol{m}_{\boldsymbol{\lambda}})) \odot \boldsymbol{s}_{\boldsymbol{\lambda}} + 1/\boldsymbol{s}_{\boldsymbol{\lambda}}. \tag{24}$$

We use $\tilde{\boldsymbol{g}}_{\boldsymbol{\lambda}}$ (along with its expectation) as a control variate to reduce the variance of the RGE $\hat{\boldsymbol{g}}_{\boldsymbol{\lambda}}$.

## 3.2   Reduced Variance Reparameterization Gradient Estimators

Now that we have constructed a tractable gradient approximation, $\tilde{\boldsymbol{g}}_{\boldsymbol{\lambda}}$, with high correlation to the original reparameterization gradient estimator, $\hat{\boldsymbol{g}}_{\boldsymbol{\lambda}}$, we can use it as a control variate as in Eq. (6)

$$\hat{\boldsymbol{g}}_{\boldsymbol{\lambda}}^{(RV)} = \hat{\boldsymbol{g}}_{\boldsymbol{\lambda}} - \boldsymbol{C}(\tilde{\boldsymbol{g}}_{\boldsymbol{\lambda}} - \mathbb{E}[\tilde{\boldsymbol{g}}_{\boldsymbol{\lambda}}]). \tag{25}$$

The optimal value for $\boldsymbol{C}$ is related to the covariance between $\tilde{\boldsymbol{g}}_{\boldsymbol{\lambda}}$ and $\hat{\boldsymbol{g}}_{\boldsymbol{\lambda}}$ (see Appendix A). We can try to estimate the value of $\boldsymbol{C}$ (or a diagonal approximation to $\boldsymbol{C}$) on the fly, or we can simply fix this value. In our case, because we are using an accurate linear approximation to the transformation of a spherical Gaussian, the optimal value of $\boldsymbol{C}$ will be close to the identity (see Appendix A.1).

**High Dimensional Models** For models with high dimensional posteriors, direct manipulation of the Hessian is computationally intractable. However, our approximations in Eqs. (22) and (23) only require a Hessian-vector product, which can be computed nearly as efficiently as the gradient [21]. Modern automatic differentiation packages enable easy and efficient implementation of Hessian-vector products for nearly any differentiable model [1, 20, 15]. We note that the mean of the control variate $\tilde{\boldsymbol{g}}_{s_{\boldsymbol{\lambda}}}$ (Eq. (24)), depends on the diagonal of the Hessian matrix. While computing the Hessian diagonal may be tractable in some cases, in general it may cost the time equivalent of $D$ function evaluations to compute [16]. Given a high dimensional problem, we can avoid this bottleneck in multiple ways. The first is simply to ignore the random variation in the Jacobian term due to $\epsilon$ — if we fix $\boldsymbol{z}$ to be $\boldsymbol{m}_{\boldsymbol{\lambda}}$ (as we do with the data term), the portion of the Jacobian that corresponds to

$s_\lambda$ will be zero (in Eq. (16)). This will result in the same Hessian-vector-product-based estimator for $\tilde{g}_{m_\lambda}$ but will set $\tilde{g}_{s_\lambda} = 0$, yielding variance reduction for the mean parameter but not the scale.

Alternatively, we can estimate the Hessian diagonal on the fly. If we use $L > 1$ samples at each iteration, we can create a per-sample estimate of the $s_\lambda$-scaled diagonal of the Hessian using the other samples [3]. As the scaled diagonal estimator is unbiased, we can construct an unbiased estimate of the control variate mean to use in lieu of the actual mean. We will see that the resulting variance is not much higher than when using full Hessian information, and is computationally tractable to deploy on high-dimensional models. A similar *local baseline* strategy is used for variance reduction in Mnih and Rezende [17].

**RV-RGE Estimators**   We introduce three different estimators based on variations of the gradient approximation defined in Eqs. (22), (23), and (24), each adressing the Hessian operations differently:

- The *Full Hessian* estimator implements the three equations as written and can be used when it is computationally feasible to use the full Hessian.
- The *Hessian Diagonal* estimator replaces the Hessian in (22) with a diagonal approximation, useful for models with a cheap Hessian diagonal.
- The *Hessian-vector product + local approximation* (HVP+Local) uses an efficient Hessian-vector product in Eqs. (22) and (23), while approximating the diagonal term in Eq. (24) using a local baseline. The HVP+Local approximation is geared toward models where Hessian-vector products can be computed, but the exact diagonal of the Hessian cannot.

We detail the RV-RGE procedure in Algorithm 1 and compare properties of these three estimators to the pure Monte Carlo estimator in the following section.

## 3.3   Related Work

Recently, Roeder et al. [27] introduced a variance reduction technique for reparameterization gradients that ignores the *parameter score* component of the gradient and can be viewed as a type of control variate for the gradient throughout the optimization procedure. This approach is complementary to our method — our approximation is typically more accurate near the beginning of the optimization procedure, whereas the estimator in Roeder et al. [27] is low-variance near convergence. We hope to incorporate information from both control variates in future work. Per-sample estimators in a multi-sample setting for variational inference were used in Mnih and Rezende [17]. We employ this technique in a different way; we use it to estimate computationally intractable quantities needed to keep the gradient estimator unbiased. Black box variational inference used control variates and Rao-Blackwellization to reduce the variance of score-function estimators [22]. Our development of variance reduction for reparameterization gradients complements their work. Other variance reduction techniques for stochastic gradient descent have focused on stochasticity due to data subsampling [12, 31]. Johnson and Zhang [12] cache statistics about the entire dataset at each epoch to use as a control variate for noisy mini-batch gradients.

The variance reduction method described in Paisley et al. [19] is conceptually similar to ours. This method uses first or second order derivative information to reduce the variance of the score function estimator. The score function estimator (and their reduced variance version) often has much higher variance than the reparameterization gradient estimator that we improve upon in this work. Our variance measurement experiments in Table 1 includes a comparison to the estimator featured in [19], which we found to be much higher variance than the baseline RGE.

## 4   Experiments and Analysis

In this section we empirically examine the variance properties of RV-RGEs and stochastic optimization for two real-data examples — a hierarchical Poisson GLM and a Bayesian neural network.[7]

- *Hierarchical Poisson GLM*: The `frisk` model is a hierarchical Poisson GLM, described in Appendix C.1. This non-conjugate model has a $D = 37$ dimensional posterior.
- *Bayesian Neural Network*: The non-conjugate `bnn` model is a Bayesian neural network applied to the `wine` dataset, (see Appendix C.2) and has a $D = 653$ dimensional posterior.

Table 1: Comparison of variances for RV-RGEs with $L = 10$-sample estimators. Variance measurements were taken for $\boldsymbol{\lambda}$ values at three points during the optimization algorithm (early, mid, late). The parenthetical rows labeled "MC abs" denote the absolute value of the standard Monte Carlo reparameterization gradient estimator. The other rows compare estimators relative to the pure MC RGE variance — a value of 100 indicates equal variation $L = 10$ samples, a value of 1 indicates a 100-fold decrease in variance (lower is better). Our new estimators (Full Hessian, Hessian Diag, HVP+Local) are described in Section 3.2. The Score Delta method is the gradient estimator described in [19]. Additional variance measurement results are in Appendix D.

| Iteration | Estimator | $g_{m_\lambda}$ | | $\ln g_{s_\lambda}$ | | $g_\lambda$ | |
|---|---|---|---|---|---|---|---|
| | | Ave $\mathbb{V}(\cdot)$ | $\mathbb{V}(\|\cdot\|)$ | Ave $\mathbb{V}(\cdot)$ | $\mathbb{V}(\|\cdot\|)$ | Ave $\mathbb{V}(\cdot)$ | $\mathbb{V}(\|\cdot\|)$ |
| early | (MC abs.) | (1.7e+02) | (5.4e+03) | (3e+04) | (2e+05) | (1.5e+04) | (5.9e+03) |
| | MC | 100.000 | 100.000 | 100.000 | 100.000 | 100.000 | 100.000 |
| | Full Hessian | 1.279 | 1.139 | 0.001 | 0.002 | 0.008 | 1.039 |
| | Hessian Diag | 34.691 | 23.764 | 0.003 | 0.012 | 0.194 | 21.684 |
| | HVP + Local | 1.279 | 1.139 | 0.013 | 0.039 | 0.020 | 1.037 |
| | Score Delta [19] | 6069.668 | 718.430 | 1.395 | 0.931 | 34.703 | 655.105 |
| mid | (MC abs.) | (3.8e+03) | (1.3e+05) | (18) | (3.3e+02) | (1.9e+03) | (1.3e+05) |
| | MC | 100.000 | 100.000 | 100.000 | 100.000 | 100.000 | 100.000 |
| | Full Hessian | 0.075 | 0.068 | 0.113 | 0.143 | 0.076 | 0.068 |
| | Hessian Diag | 38.891 | 21.283 | 6.295 | 7.480 | 38.740 | 21.260 |
| | HVP + Local | 0.075 | 0.068 | 30.754 | 39.156 | 0.218 | 0.071 |
| | Score Delta [19] | 4763.246 | 523.175 | 2716.038 | 700.100 | 4753.752 | 523.532 |
| late | (MC abs.) | (1.7e+03) | (1.3e+04) | (1.1) | (19) | (8.3e+02) | (1.3e+04) |
| | MC | 100.000 | 100.000 | 100.000 | 100.000 | 100.000 | 100.000 |
| | Full Hessian | 0.042 | 0.030 | 1.686 | 0.431 | 0.043 | 0.030 |
| | Hessian Diag | 40.292 | 53.922 | 23.644 | 28.024 | 40.281 | 53.777 |
| | HVP + Local | 0.042 | 0.030 | 98.523 | 99.811 | 0.110 | 0.022 |
| | Score Delta [19] | 5183.885 | 1757.209 | 17355.120 | 3084.940 | 5192.270 | 1761.317 |

**Quantifying Gradient Variance Reduction** We measure the variance reduction of the RGE observed at various iterates, $\boldsymbol{\lambda}_t$, during execution of gradient descent. Both the gradient magnitude, and the marginal variance of the gradient elements — using a sample of 1000 gradients — are reported. Further, we inspect both the mean, $\boldsymbol{m_\lambda}$, and log-scale, $\ln \boldsymbol{s_\lambda}$, parameters separately. Table 1 compares gradient variances for the `frisk` model for our four estimators: i) pure Monte Carlo (MC), ii) Full Hessian, iii) Hessian Diagonal, and iv) Hessian-vector product + local approximation (HVP+Local). Additionally, we compare our methods to the estimator described in [19], based on the score function estimator and a control variate method. We use a first order delta method approximation of the model term, which admits a closed form control variate term.

Each entry in the table measures the percent of the variance of the pure Monte Carlo estimator. We show the average variance over each component Ave$\mathbb{V}(\cdot)$, and the variance of the norm $\mathbb{V}(\|\cdot\|)$. We separate out variance in mean parameters, $\boldsymbol{g_m}$, log scale parameters, $\ln \boldsymbol{g_s}$, and the entire vector $\boldsymbol{g_\lambda}$. The reduction in variance is dramatic. Using HVP+Local, in the norm of the mean parameters we see between a $80\times$ and $3{,}000\times$ reduction in variance depending on the progress of the optimizer. The importance of the full Hessian-vector product for reducing mean parameter variance is also demonstrated as the Hessian diagonal only reduces mean parameter variance by a factor of 2-5$\times$.

For the variational scale parameters, $\ln \boldsymbol{g_s}$, we see that early on the HVP+Local approximation is able to reduce parameter variance by a large factor ($\approx 2{,}000\times$). However, at later iterates the HVP+Local scale parameter variance is on par with the Monte Carlo estimator, while the full Hessian estimator still enjoys huge variance reduction. This indicates that, by this point, most of the noise is the local Hessian diagonal estimator. We also note that in this problem, most of the estimator variance is in the mean parameters. Because of this, the norm of the entire parameter gradient, $\boldsymbol{g_\lambda}$ is reduced by $100 - 5{,}000\times$. We found that the score function estimator (with the delta method control variate) is typically much higher variance than the baseline reparameterization gradient estimator (often by a factor of 10-50$\times$). In Appendix D we report results for other values of $L$.

**Optimizer Convergence and Stability** We compare the optimization traces for the `frisk` and `bnn` model for the MC and the HVP+Local estimators under various conditions. At each iteration we estimate the true ELBO value using 2000 Monte Carlo samples. We optimize the ELBO objective using `adam` [13] for two step sizes, each trace starting at the same value of $\boldsymbol{\lambda}_0$.

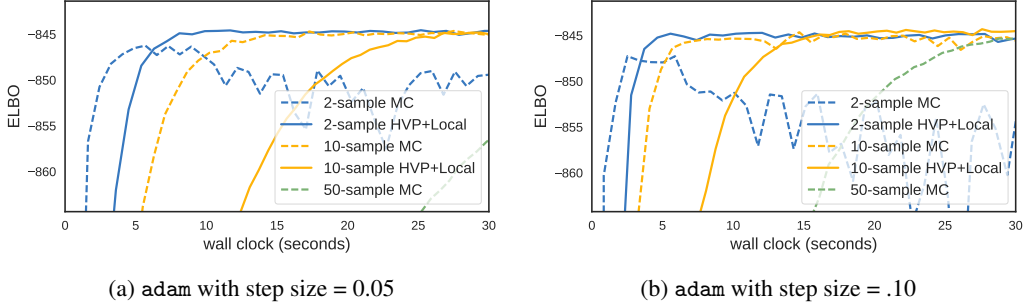

(a) `adam` with step size = 0.05        (b) `adam` with step size = .10

Figure 3: MCVI optimization trace applied to the `frisk` model for two values of $L$ and step size. We run the standard MC gradient estimator (solid line) and the RV-RGE with $L = 2$ and $10$ samples.

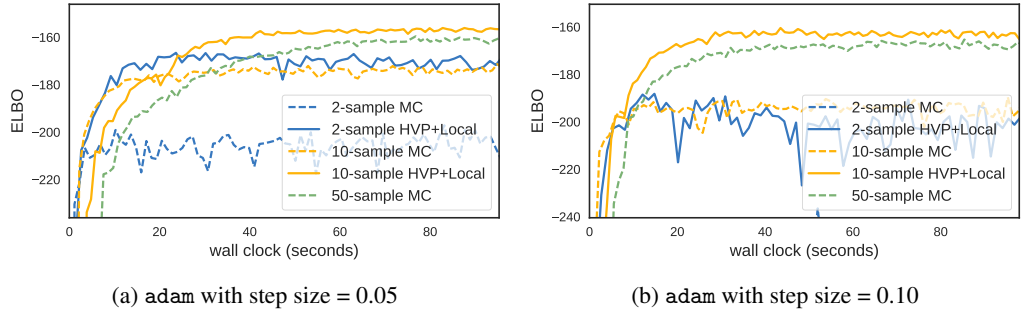

(a) `adam` with step size = 0.05        (b) `adam` with step size = 0.10

Figure 4: MCVI optimization for the `bnn` model applied to the `wine` data for various $L$ and step sizes. The standard MC gradient estimator (dotted) was run with 2, 10, and 50 samples; RV-RGE (solid) was run with 2 and 10 samples. In 4b the 2-sample MC estimator falls below the frame.

Figure 3 compares ELBO optimization traces for $L = 2$ and $L = 10$ samples and step sizes .05 and .1 for the `frisk` model. We see that the HVP+Local estimators make early progress and converge quickly. We also see that the $L = 2$ pure MC estimator results in noisy optimization paths. Figure 4 shows objective value as a function of wall clock time under various settings for the `bnn` model. The HVP+Local estimator does more work per iteration, however it tends to converge faster. We observe the $L = 10$ HVP+Local outperforming the $L = 50$ MC estimator.

## 5 Conclusion

Variational inference reframes an integration problem as an optimization problem with the caveat that each step of the optimization procedure solves an easier integration problem. For general models, each sub-integration problem is itself intractable, and must be estimated, typically with Monte Carlo samples. Our work has shown that we can use more information about the variational family to create tighter estimators of the ELBO gradient, which leads to faster and more stable optimization. The efficacy of our approach relies on the complexity of the RGE distribution to be well-captured by linear structure which may not be true for all models. However, we found the idea effective for non-conjugate hierarchical Bayesian models and a neural network.

Our presentation is a specific instantiation of a more general idea — using cheap linear structure to remove variation from stochastic gradient estimates. This method described in this work is tailored to Gaussian approximating families for Monte Carlo variational inference, but could be easily extended to location-scale families. We plan to extend this idea to more flexible variational distributions, including flow distributions [24] and hierarchical distributions [23], which would require approximating different functional forms within the variational objective. We also plan to adapt our technique to model and inference schemes with recognition networks [14], which would require back-propagating de-noised gradients into the parameters of an inference network.

**Acknowledgements**

The authors would like to thank Finale Doshi-Velez, Mike Hughes, Taylor Killian, Andrew Ross, and Matt Hoffman for helpful conversations and comments on this work. ACM is supported by the Applied Mathematics Program within the Office of Science Advanced Scientific Computing Research of the U.S. Department of Energy under contract No. DE-AC02-05CH11231. NJF is supported by a Washington Research Foundation Innovation Postdoctoral Fellowship in Neuroengineering and Data Science. RPA is supported by NSF IIS-1421780 and the Alfred P. Sloan Foundation.

## Footnotes

[2]We use $q(\boldsymbol{z}; \boldsymbol{\lambda})$ and $q_{\boldsymbol{\lambda}}(\boldsymbol{z})$ interchangeably.

[3]We will also use $x/y$ and $x^2$ to denote pointwise division and squaring, respectively.

[4]We require the model $\ln p(\boldsymbol{z}, \mathcal{D})$ to be twice differentiable.

[5]In fact, we know that the expectation of the *parameter score* term is zero, and removing that term altogether can sometimes be a source of variance reduction that we do not explore here [27].

[6]For diagonal Gaussian $q$, we define $\boldsymbol{\lambda} = [\boldsymbol{m}_{\boldsymbol{\lambda}}, \boldsymbol{s}_{\boldsymbol{\lambda}}]$.

[7]Code is available at `https://github.com/andymiller/ReducedVarianceReparamGradients`.

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
