[Supplementary Material]

# A Control Variates

Control variates are random quantities that are used to reduce the variance of a statistical estimator without trading any bias. Concretely, given an unbiased estimator $\hat{g}$ such that $\mathbb{E}[\hat{g}] = g$ (the quantity of interest), our goal is to construct another unbiased estimator with lower variance. We can do this by defining a *control variate* $\tilde{g}$ with *known expectation* $\tilde{m}$. We can write our new estimator as

$$g^{(cv)} = \hat{g} - c \cdot (\tilde{g} - \tilde{m}). \tag{26}$$

Clearly the new estimator has the same expectation as the original estimator, but a different variance. We can reduce the variance of $g^{(cv)}$ by setting $c$ optimally.

Consider a univariate $\hat{g}$ and $\tilde{g}$, and without loss of generality, take $\tilde{m} = 0$. The variance of $g^{(cv)}$ can be written

$$\mathbb{V}(g^{(cv)}) = \mathbb{E}[(\hat{g} - c \cdot \tilde{g})^2] - \mathbb{E}[\hat{g}]^2 \tag{27}$$

$$= \mathbb{E}[\hat{g}^2 + c^2 \cdot \tilde{g}^2 - 2c\hat{g}\tilde{g}] - \mathbb{E}[\hat{g}]^2 \tag{28}$$

$$= \mathbb{E}[\hat{g}^2] + c^2 \mathbb{E}[\tilde{g}^2] - 2c\mathbb{E}[\hat{g}\tilde{g}] - \mathbb{E}[\hat{g}]^2 \tag{29}$$

We minimize the variance with respect to $c$ by taking the derivative and setting equal to zero, which implies

$$c^* = \frac{\mathbb{E}[\hat{g}\tilde{g}]}{\mathbb{E}[\tilde{g}^2]} = \frac{\mathbb{C}(\hat{g}, \tilde{g})}{\mathbb{V}(\tilde{g})} \tag{30}$$

The covariance $\mathbb{C}(\hat{g}, \tilde{g})$ is typically not known a priori and must be estimated. It can be shown, under the optimal $c^*$, that the variance of $g^{(cv)}$ is

$$\mathbb{V}(g^{(cv)}) = (1 - \rho^2)\mathbb{V}(\hat{g}) \tag{31}$$

where $\rho$ is the correlation coefficient between $\tilde{g}$ and $\hat{g}$.

When $\hat{g}$ and $\tilde{g}$ are length $D$ vectors, we can construct an estimator that depends on a matrix-valued free parameter, $C \in \mathbb{R}^{D \times D}$

$$g^{(cv)} = \hat{g} - C(\tilde{g} - \tilde{m}). \tag{32}$$

We can show that the $C$ that minimizes the $\text{Tr}(\mathbb{C}(g^{(cv)}))$ — the sum of the marginal variances — is given by

$$C^* = \Sigma_{\tilde{g}}^{-1} \Sigma_{\hat{g},\tilde{g}} \tag{33}$$

where $\Sigma_{\tilde{g}}$ is the covariance matrix of the control variate vector, and $\Sigma_{\hat{g},\tilde{g}}$ is the cross covariance between $\hat{g}$ and $\tilde{g}$.

Intuitively, a control variate is injecting information into the estimator in the form of linear structure. If the two quantities, $\tilde{g}$ and $\hat{g}$ are perfectly correlated, then we already know the mean and estimation is not necessary. As the two become uncorrelated, the linear estimator becomes less and less informative, and reverts to the original quantity.

## A.1 Control Variates and Approximate Functions

In our setting, we approximate the distribution of some function $f(\epsilon)$ where $\epsilon \sim \mathcal{N}(0, I)$ by a first order Taylor expansion about 0 — for now we examine the univariate case

$$f_1(\epsilon) = f(0) + f'(0)\epsilon \quad \epsilon \in \mathbb{R} \tag{34}$$

If we wish to use $f_1(\epsilon)$ as a control variate for $f(\epsilon)$, we need to characterize the covariance between the two random variables. Because the form of $f(\epsilon)$ is general, it is difficult to analyze. We instead derive the covariance between $f_1(\epsilon)$ and the second-order expansion

$$f_2(\epsilon) = f(0) + f'(0)\epsilon + f''(0)/2\epsilon^2 \tag{35}$$

as a surrogate.

$$\mathbb{C}(\boldsymbol{f}_1(\epsilon), \boldsymbol{f}_2(\epsilon)) = \mathbb{E}\left[(\boldsymbol{f}_1(\epsilon) - \mathbb{E}[\boldsymbol{f}_1(\epsilon)])(\boldsymbol{f}_2(\epsilon) - \mathbb{E}[\boldsymbol{f}_2(\epsilon)])\right] \tag{36}$$

$$= \mathbb{E}\left[(\boldsymbol{f}'(0)\epsilon)\left(\boldsymbol{f}'(0)\epsilon + \boldsymbol{f}''(0)/2\epsilon^2 - \boldsymbol{f}''(0)/2\right)\right] \tag{37}$$

$$= \mathbb{E}\left[\boldsymbol{f}'(0)^2\epsilon^2 + (\boldsymbol{f}'(0)\boldsymbol{f}''(0)/2)\epsilon^3 - (\boldsymbol{f}'(0)\boldsymbol{f}''(0)/2)\epsilon\right] \tag{38}$$

$$= \mathbb{E}\left[\boldsymbol{f}'(0)^2\epsilon^2\right] \tag{39}$$

$$= \mathbb{V}[\boldsymbol{f}_1(\epsilon)] \tag{40}$$

where note that $\mathbb{E}[\epsilon^3] = 0$. Recall that the optimal control variate can be written

$$c^* = \mathbb{C}(\boldsymbol{f}_1(\epsilon), \boldsymbol{f}_2(\epsilon))/\mathbb{V}[\boldsymbol{f}_1(\epsilon)] \tag{41}$$

$$= \mathbb{V}[\boldsymbol{f}_1(\epsilon)]/\mathbb{V}[\boldsymbol{f}_1(\epsilon)] = 1 . \tag{42}$$

## B   Algorithm Details

We summarize an optimization routine using RV-RGE in Algorithm 1. The different variants rely on the different forms of $\boldsymbol{H}(\cdot, \cdot)$ and $\mathrm{diag}(\boldsymbol{H})$. The *full Hessian* estimator calculates these terms exactly. The *diagonal Hessian* estimates the Hessian-vector product with the diagonal of the Hessian. The *HVP+Local* estimator computes the Hessian-vector product exactly, but estimates the scale approximation mean using other samples.

We also note that there are ways to optimize the additional Hessian-vector product computation. Because each Hessian is evaluated at the same $\boldsymbol{m}_\lambda$, we can cache the computation in the forward pass, and only repeat the backwards pass for each sample, as implemented in [15].

## C   Model Definitions

### C.1   Multi-level Poisson GLM

Our second test model is a 37-dimensional posterior resulting from a hierarchical Poisson GLM. This model measures the relative rates of stop-and-frisk events for different ethnicities and in different precincts [6], and has been used as illustrative example of multi-level modeling [5, Chapter 15, Section 1].

$$
\begin{aligned}
\mu &\sim \mathcal{N}(0, 10^2) && \text{mean offset} \\
\ln \sigma_\alpha^2, \ln \sigma_\beta^2 &\sim \mathcal{N}(0, 10^2) && \text{group variances} \\
\alpha_e &\sim \mathcal{N}(0, \sigma_\alpha^2) && \text{ethnicity effect} \\
\beta_p &\sim \mathcal{N}(0, \sigma_\beta^2) && \text{precinct effect} \\
\ln \lambda_{ep} &= \mu + \alpha_e + \beta_p + \ln N_{ep} && \text{log rate} \\
Y_{ep} &\sim \mathcal{P}(\lambda_{ep}) && \text{stop-and-frisk events}
\end{aligned}
$$

where $Y_{ep}$ are the number of stop-and-frisk events within ethnicity group $e$ and precinct $p$ over some fixed period of time; $N_{ep}$ is the total number of arrests of ethnicity group $e$ in precinct $p$ over the same period of time; $\alpha_e$ and $\beta_p$ are the ethnicity and precinct effects.

### C.2   Bayesian Neural Network

We implement a 50-unit hidden layer neural network with ReLU activation functions. We place a normal prior over each weight in the neural network, governed by the same variance (with an inverse Gamma prior). We also place an inverse Gamma prior over the observation variance The model can be written as

$$
\begin{aligned}
\alpha &\sim \mathrm{Gamma}(1, .1) && \text{weight prior hyper} & (43) \\
\tau &\sim \mathrm{Gamma}(1, .1) && \text{noise prior hyper} & (44) \\
w_i &\sim \mathcal{N}(0, 1/\alpha) && \text{weights} & (45) \\
y|x, w, \tau &\sim \mathcal{N}(\phi(x, w), 1/\tau) && \text{output distribution} & (46)
\end{aligned}
$$

where $w = \{w\}$ is the set of weights, and $\phi(x, w)$ is a multi-layer perceptron that maps input $x$ to approximate output $y$ as a function of parameters $w$. We denote the set of parameters as $\theta \triangleq (w, \alpha, \tau)$. We approximate the posterior $p(w, \alpha, \tau | \mathcal{D})$, where $\mathcal{D}$ is the training set of $\{x_n, y_n\}_{n=1}^N$ input-output pairs.

We use a 100-row subsample of the `wine` dataset from the UCI repository `https://archive.ics.uci.edu/ml/datasets/Wine+Quality`.

## D  Variance Reduction

Below are additional variance reduction measurements for the `frisk` model for different values of $L$, samples drawn per iteration. We measure the variance of the variational parameter gradient at three points during the optimization procedure: (i) early, near initialization, (ii) mid, before convergence, (iii) late, near convergence. We compare four methods

- MC: Monte Carlo estimator using the reparameterization trick
- Full Hessian: Our reduced variance gradient using the full hessian calculation
- Hessian Diag: Our reduced variance gradient using only diagonal Hessian information
- HVP + Local: Our fast reduced variance gradient estimator, using only Hessian-vector products and a local baseline
- Score Delta: Method described in [19] using a control variate with the score function estimator of the gradient.

Table 2: `frisk` model variance comparison: $L = 3$-sample estimators

| Iteration | Estimator | $g_{m_\lambda}$ Ave $\mathbb{V}(\cdot)$ | $\mathbb{V}(\|\|\cdot\|\|)$ | $\ln g_{s_\lambda}$ Ave $\mathbb{V}(\cdot)$ | $\mathbb{V}(\|\|\cdot\|\|)$ | $g_\lambda$ Ave $\mathbb{V}(\cdot)$ | $\mathbb{V}(\|\|\cdot\|\|)$ |
|---|---|---|---|---|---|---|---|
| early | (MC abs.) | (5.4e+02) | (1.7e+04) | (9.6e+04) | (5.9e+05) | (4.8e+04) | (1.9e+04) |
| | MC | 100.000 | 100.000 | 100.000 | 100.000 | 100.000 | 100.000 |
| | Full Hessian | 1.184 | 1.022 | 0.001 | 0.002 | 0.007 | 0.902 |
| | Hessian Diag | 35.541 | 25.012 | 0.003 | 0.011 | 0.201 | 22.090 |
| | HVP + Local | 1.184 | 1.022 | 0.012 | 0.039 | 0.019 | 0.900 |
| | Score Delta [19] | 6054.168 | 651.784 | 1.429 | 1.783 | 35.134 | 574.536 |
| mid | (MC abs.) | (1.4e+04) | (4.5e+05) | (63) | (1.1e+03) | (6.9e+03) | (4.5e+05) |
| | MC | 100.000 | 100.000 | 100.000 | 100.000 | 100.000 | 100.000 |
| | Full Hessian | 0.080 | 0.075 | 0.122 | 0.169 | 0.081 | 0.075 |
| | Hessian Diag | 39.016 | 22.832 | 6.617 | 8.097 | 38.868 | 22.804 |
| | HVP + Local | 0.080 | 0.075 | 31.992 | 46.160 | 0.227 | 0.078 |
| | Score Delta [19] | 4787.771 | 1031.561 | 2833.663 | 1619.190 | 4778.818 | 1033.613 |
| late | (MC abs.) | (5.6e+03) | (5.4e+04) | (4.1) | (74) | (2.8e+03) | (5.4e+04) |
| | MC | 100.000 | 100.000 | 100.000 | 100.000 | 100.000 | 100.000 |
| | Full Hessian | 0.044 | 0.024 | 1.782 | 0.879 | 0.045 | 0.023 |
| | Hessian Diag | 39.280 | 38.799 | 22.915 | 21.913 | 39.268 | 38.725 |
| | HVP + Local | 0.044 | 0.024 | 98.290 | 99.679 | 0.116 | 0.014 |
| | Score Delta [19] | 5019.294 | 2804.652 | 15681.050 | 5650.339 | 5027.114 | 2810.160 |

Table 3: `frisk` model variance comparison: $L = 50$-sample estimators

| Iteration | Estimator | $g_{m_\lambda}$ Ave $\mathbb{V}(\cdot)$ | $\mathbb{V}(\|\cdot\|)$ | $\ln g_{s_\lambda}$ Ave $\mathbb{V}(\cdot)$ | $\mathbb{V}(\|\cdot\|)$ | $g_\lambda$ Ave $\mathbb{V}(\cdot)$ | $\mathbb{V}(\|\cdot\|)$ |
|---|---|---|---|---|---|---|---|
| early | (MC abs.) | (34) | (1.1e+03) | (6.1e+03) | (4e+04) | (3.1e+03) | (1.1e+03) |
|  | MC | 100.000 | 100.000 | 100.000 | 100.000 | 100.000 | 100.000 |
|  | Full Hessian | 1.276 | 1.127 | 0.001 | 0.002 | 0.008 | 1.080 |
|  | Hessian Diag | 35.146 | 24.018 | 0.003 | 0.012 | 0.197 | 23.028 |
|  | HVP + Local | 1.276 | 1.127 | 0.013 | 0.039 | 0.020 | 1.079 |
|  | Score Delta [19] | 6084.473 | 765.666 | 1.384 | 0.535 | 34.957 | 734.007 |
| mid | (MC abs.) | (7.4e+02) | (2.4e+04) | (3.4) | (81) | (3.7e+02) | (2.4e+04) |
|  | MC | 100.000 | 100.000 | 100.000 | 100.000 | 100.000 | 100.000 |
|  | Full Hessian | 0.081 | 0.074 | 0.125 | 0.121 | 0.081 | 0.074 |
|  | Hessian Diag | 37.534 | 21.773 | 7.204 | 7.035 | 37.394 | 21.752 |
|  | HVP + Local | 0.081 | 0.074 | 31.278 | 32.275 | 0.225 | 0.076 |
|  | Score Delta [19] | 5115.048 | 557.946 | 3047.996 | 354.204 | 5105.546 | 557.329 |
| late | (MC abs.) | (3.3e+02) | (1.8e+03) | (0.23) | (4.4) | (1.7e+02) | (1.8e+03) |
|  | MC | 100.000 | 100.000 | 100.000 | 100.000 | 100.000 | 100.000 |
|  | Full Hessian | 0.042 | 0.043 | 1.894 | 0.296 | 0.044 | 0.043 |
|  | Hessian Diag | 39.972 | 101.263 | 24.450 | 27.174 | 39.961 | 101.019 |
|  | HVP + Local | 0.042 | 0.043 | 98.588 | 99.539 | 0.112 | 0.033 |
|  | Score Delta [19] | 5192.542 | 1422.083 | 16907.603 | 1376.037 | 5200.855 | 1424.831 |