[Reviews · NeurIPS 2017]

Reviewer 1



Summary This paper proposes a control variate (CV) for the reparametrization gradient by exploiting a linearization of the data model score. For Gaussian random variables, such a linearization has a distribution with a known mean, allowing its use as a CV. Experiments show using the CV results in faster (according to wall clock time) ELBO optimization for a GLM and Bayesian NN. Furthermore, the paper reports 100 fold (+) variance decreases during optimization of the GLM. Evaluation Method: The CV proposed is clever; the observation that the linearization of the data score has a known distribution is non-obvious and interesting. This is a contribution that can easily be incorporated when using the reparametrization trick. The only deficiencies of the method are (1) it requires the model’s Hessian and (2) application is limited to Gaussian random variables. It also does not extend to amortized inference (i.e. variational autoencoders), but this limitation is less important. I think the draft could better detail why it’s limited to Gaussian variables (which is a point on which I elaborate below under ‘Presentation’). Some discussion of the hurdles blocking extension to other distributions would improve the draft and be appreciated by readers. For instance, it looks like the linearized form might have a known distribution for other members of the location-scale family (since they would all have the form f(m) + H(m)(s*epsilon))? Thoughts on this? Experiments: Experiments are limited to small models (by modern standards)---GLM and one-layer / 50 hidden unit Bayesian NN---but are adequate to demonstrate the correctness and utility of the method. I would have liked to see what held-out performance gains can be had by using the method---you must have looked at held-out LL?---but I realize this is not crucial for assessing the method as a CV. Still I think the paper is strong enough that even a negative or neutral result would not hurt the paper’s quality. Presentation: The paper’s presentation of the method’s derivation is clear and detailed. I had no trouble seeing the logical flow from equation to equation. However, I think the draft’s description is somewhat ‘shallow’---by that I mean, while the algebra is clear, more discussion and intuition behind the equations could be included. Specifically, after the method is derived, I think a section discussing why it's limited to Gaussian rv’s and why it can’t be extended to amortized inference would be beneficial to readers. It took me two-to-three reads through the draft before I fully understood these limitations. To make room for this discussion, I would cut from the introduction and VI background since, if a person is interested in implementing your CV, they likely already understand VI, the ELBO, etc. Minutiae: Table 1 is misreferenced as ‘Table 3’ in text. Conclusion: This paper was a pleasure to read, and I recommend its acceptance. The methodology is novel and interesting, and the experiments adequately support the theory. The draft could be improved with more exposition (on limitations, for instance), but this is not a crucial flaw.

Reviewer 2



This paper shows how to build a control variate to reduce the variance of the reparameterization gradient estimator in variational inference. The main idea is to linearize the "data term" (the gradient of the log-joint) around the mean of the variational distribution, and use this linearized term to build the control variate. Results show faster convergence of the ELBO and significant variance reduction. Overall, the idea is interesting and the writing quality is good. However, I have some concerns, specially with computational complexity: 1. The proposed approach relies on the Hessian information from the model. The paper mentions efficient ways to compute Hessian-vector products (ref. [15]), but it should include further analysis on the resulting complexity of the inference for each of the approaches that are described (full Hessian, diagonal Hessian, HVP+Local). 2. Related to the point above, the experiments are done on relatively small models (D=37 and D=653). That suggests that scalability is an issue with this method. The paper would be significantly improved with experiments on more realistic scenarios, with at least thousands or tens of thousands latent variables. 3. Table I should also include vanilla reparameterization gradients (i.e., with no control variates), not only comparisons of methods described in the paper. 4. The paper focuses on the Gaussian case. This is sensible because it is the most common use for reparameterization gradients. I wonder if the proposed approach can be extended beyond the Gaussian case, as it would become harder to fully define the distribution of tilde{g}, similarly to Eqs. 21-22. The papers states (lines 124, 138) that for some transformations T we can compute this distribution; is there any other example besides the Gaussian? Here are also some minor comments: 1. The discussion about high-dimensional models (lines 176-186) is not clear to me. 2. In line 239, it reads "as the Hessian diagonal only reduces mean parameter variance by a factor of 2-5". That doesn't seem to be the case according to Table 1. 3. In line 157, the text reads "The data term [...] is all that remains to be approximated". This is misleading, as it seems to imply that the Eqs. before (Eqs. (15)-(18)) are approximations, when they are not. 4. I think that "Doubly Stochastic Variational Bayes for non-Conjugate Inference", which was developed in parallel to [19], should be cited in line 36. 5. Eqs. (15)-(18) are standard in reparameterization trick for VI; I think these can be moved to the appendix to gain some space. 6. There are some typos: "parameter" (151), "complements" (216), "Table 1" (230), "L=2" (253).

Reviewer 3



I have read the author feedback and have adjusted my rating and comments accordingly. Summary: The paper proposes a control variate to reduce the variance of the reparameterisation gradients in Monte Carlo variational inference. The reparameterisation gradient can be decomposed into several terms and a local linearisation can be done on the data term of this gradient. This observation results in a tractable control variate which relates the structure of the variational approximation to the curvature of the unapproximated joint density. The experiment results seem to support this control variate on two models [a generalised linear model and a small Bayesian neural network] that use Gaussian variational approximations. Details: I enjoyed reading this paper -- it was clearly written and the structure is well-thought-out and easy to follow/understand. It would be great if alternative control variates are discussed and compared against, for example, the delta control variate technique in Paisley et al.'s stochastic search for VI paper. One disadvantage of this approach is that it requires the Hessian or Hessian vector product of the joint density of the model. These are expensive to evaluate in high dimensions [additionally, these involve a sum over the training instances]. Even though (online or diagonal or monte carlo) approximations are available they don't seem to help reduce variances and even hurt the performance. My personal opinion is that more cheap and cheerful, and *effective* approximations are needed for this method to be deployed practice to more realistic cases (bigger models + bigger datasets). Would it help to decompose the joint density further into a sum of prior density + log likelihood, then check the gradients given by these extra terms and perform linearisation only on the log likelihood instead of on the joint?